# Predictors and Importance of Social Aspects in Ikigai among Older Women

**DOI:** 10.3390/ijerph18168718

**Published:** 2021-08-18

**Authors:** Kazuki Seko, Michiyo Hirano

**Affiliations:** 1Graduate School of Health Sciences, Hokkaido University, Sapporo 060-0812, Hokkaido, Japan; ks.the.hto2@gmail.com; 2Faculty of Health Sciences, Hokkaido University, Sapporo 060-0812, Hokkaido, Japan

**Keywords:** successful aging, women, social activity

## Abstract

Ikigai is a unique Japanese concept that encompasses not only joy and happiness, but also the purpose and meaning of life. The purpose of this study was to explore the factors that contribute to Ikigai, an important concept in the health of older women, and to clarify the relationship between Ikigai and the enhancement of the social aspects of their health. The participants in this longitudinal study, conducted between October 2017 and February 2020, were physically healthy older women aged 65 years and above (*N* = 132). Physical function and social activity were used as predictors of Ikigai. In addition, willingness for new interactions was used to examine the social aspects of health. A causal model was created from these factors, and path analysis was performed. The results demonstrated that participating in numerous interpersonal activities was predictive of increased Ikigai, but physical function was not. In addition, Ikigai was found to affect the willingness for new interactions. Further, Ikigai could be related to the willingness for new interactions. To enhance the social aspects of older women’s health, it is necessary to focus on Ikigai.

## 1. Introduction

Older adults experience many losses that they must cope with, including health, social, economic, and relationship losses [1]. Havighurst describes the developmental tasks of old age as adjusting to declining strength and health, retirement and loss of income, and having intimate relationships with people of the same generation [2], among other things. Adapting to these old-age-related losses is an important challenge for older adults.

In addition, successful aging involves avoiding disease and disability, having high cognitive and physical functioning, as well as engaging with life, such as having interpersonal relationships and social contributions [1]. In other words, in addition to physical and mental health, such as physical functioning and the presence or absence of disease, the fulfillment of social aspects, such as interpersonal connections, is necessary for a better life in old age. In particular, women attach more value to developing close interpersonal relations than men [3], and the link between social aspects and health is stronger for older women than older men [4,5,6]. Therefore, the fulfillment of social aspects, such as interacting with people of the same generation, is an issue that needs additional attention among older women.

This study focused on Ikigai, which is a concept considered to contribute to social fulfillment in older women. Ikigai, the notion of a life worth living, is commonly used as a subjective measure of well-being. However, as a comprehensive concept that includes joy and happiness, as well as the purpose and meaning of life, it is more than that [7,8,9]. There is a link between Ikigai and health: having Ikigai has been shown to reduce the risk of heart and other diseases as well as the risk of associated mortality [10,11,12]. Ikigai is a Japanese concept; however, it is considered synonymous with purpose in life (PIL) [13]. PIL studies confirm the association between PIL and risk of death [14]. Further, both Ikigai and PIL are recognized as having similar effects on physical health. As Ikigai is easily lost with age [15], it is necessary to explore the factors that contribute to high Ikigai and to clarify the relationship between Ikigai and social fulfillment.

Ikigai is predicted to be associated with strong interpersonal ties [16]. Positive perceptions of the future and social satisfaction are unique aspects of Ikigai [15]. Acquiring Ikigai has many positive physical effects, which converge on the individual, and may also include social effects, such as relationships with others. In other words, having Ikigai may lead to a positive attitude toward the enrichment of relationships, which will, in turn, enhance interpersonal connections. Social relationships are also central to the related concept of PIL [17]; therefore, the effects of Ikigai and PIL on the social aspects of health may be similar.

In Japan, social isolation of older people, especially women, is an important issue that needs to be addressed. Of its total population, 28.1% is over 65 years old, and this percentage has increased by more than 10% in the past 20 years [18]. In this situation, social isolation of older people is becoming a serious issue in Japan, especially among women [19]. Among older women, isolation is an important factor that leads to depression [20]. Positive emotions can be considered an effective way for older people to deal with this problem. According to the “broaden and build theory”, positive emotions help individuals to become more socially integrated, healthier, and better off [21]. In the West, the emphasis for happiness is on achieving personal goals, while in Asia, including Japan, the emphasis is on achieving mutual goals with others [22]. This is why, among positive emotions, Ikigai, which reflects social satisfaction, is so important for women in Japan.

This study focused on the physical function and social activities of aging adults as factors that enhance their Ikigai. The former is older adults’ basic ability that is essential to their daily living [23]. High physical function is associated with the amount of activity older people can engage in [24], which increases quality of life and life satisfaction [25,26,27]. High levels of physical functioning are a prerequisite for being active in old age; having this ability can enable older people to lead a fulfilling life by performing activities and gaining various experiences. A high level of physical functioning is predicted to be important for them to be able to view their lives as meaningful and to have Ikigai.

Participation in numerous social activities contributes to older adults’ health [28]. Older women tend to be more involved in the community; consequently, the impact of social activity on their well-being is more pronounced [29,30,31]. Social activities are instances where they can relate to others and, for those who value interpersonal relationships, these activities can provide them with a sense of meaning in their lives [3,32]. It is predicted that this will increase their Ikigai.

Thus far, no studies have addressed factors that increase Ikigai in older women or focused on the effect it has on the social aspects of health. The purpose of this research was twofold: first, to identify the factors that contribute to high Ikigai among older women, and second, to quantitatively assess whether Ikigai affects their willingness to engage in new interactions with people of the same generation (henceforth referred to as “willingness for new interactions”). The following hypotheses were formulated:(1)Ikigai in older women is affected by their physical function and social activities, and(2)Ikigai in older women affects their willingness for new interactions.

## 2. Materials and Methods

### 2.1. Study Population

The target sample comprised older women aged 65 years and above, attending the “Care prevention group activities” in Japan that combine exercise, recreation, and tea parties to facilitate socialization among individuals to prevent falls and dementia [33]. The criterion for health was that the patient must not be certified as having “accidental functional disability,”, a newly recognized official classification of Japan’s long-term care insurance (LTCI) system [34]. The LTCI accreditation process consists of an initial assessment, followed by scoring and reassessment. Trained local government officials assess care needs using a questionnaire about current physical and mental condition and the use of medical procedures. There is an initial computer-based assessment, and then a reassessment by a specialist. The reassessment is conducted by the Care Needs Certification Board, which consists of physicians, nurses, and other health and social work professionals. Subjects are classified into one of seven levels, or none, depending on their care needs [34].

In this study, those who were certified by the LTCI into one of the seven levels before or during the study period, that is, those who were certified as needing nursing care from others, were excluded from the analysis. The level of LTCI certification has been shown to be closely related to the ability to perform activities of daily living [35].

### 2.2. Research Design and Data Collection

A two-year longitudinal survey was conducted on the target population; a baseline and a follow-up survey were conducted in 2017 and 2019, respectively. The survey period for the former was from October 2017 to March 2018, while that for the latter was from September 2019 to February 2020. This survey employed a collective method and was carried out using a self-administered written questionnaire. The process of participant selection is shown in Figure 1. A total of 417 individuals participated in the baseline survey, of which 84 were males or LTCI-certified. Of this total, 140 continued to participate in the follow-up survey, of which 132 had complete data from the study measures and were thus included in the analysis.

### 2.3. Ethical Considerations

The participants were provided with written and verbal information about the study’s purpose and methods, the processing of responses, and ethical considerations. Informed consent was obtained from all the study’s participants. Additionally, they were informed that they could withdraw their consent at any time of their own will. Their participation in the research was voluntary, anonymity was guaranteed, and the data collected were password protected and strictly controlled. The Ethics Review Committee of the Faculty of Health Sciences, Hokkaido University approved this study (no. 17–80, 11 October 2017; no. 19–52, 26 August 2019).

### 2.4. Measures

The questions collected information on basic attributes, social activities, Ikigai, and willingness for new interactions. Additionally, all participants provided a measure of grip strength as an indicator of physical function. Each participant filled out a questionnaire with these items.

Personal Characteristics. These were measured based on age, residence status, and subjective health perception. The subjective health item was “Please rate your current state of health as you perceive it” that was responded to using one of the following four alternatives: “1. Not healthy”, “2. Not so healthy”, “3. Slightly healthy”, and “4. Very healthy.”

Social Activity. To assess this dimension, we asked the question, “Please circle the number that applies to your participation in each of the following activities.” Participants responded dichotomously with either “Yes” or “No”, which were scored as 1 and 0, respectively. The activities were: “Neighborhood association activities”, “Older people’s club”, “Salon”, “Volunteer activities”, “Study sessions”, “Sports”, “Hobby/entertainment”, and “Travel/visit”. For social activities, we used items from the Cabinet Office’s “Survey on the Economic and Living Environment of the Elderly” [36] and the Ministry of Internal Affairs and Communications’ “Basic Survey on Social Life” [37]. These are used to survey the status of older people’s social activities. Furthermore, to accurately reflect the status of social activities conducted by the survey’s participants, the items included “Salon” and “Older people’s club” activities, which are activities held in the areas where the survey was conducted. Salons are social gatherings held by local governments and citizen volunteers for older adults. Individuals can engage in various activities such as dancing and socializing, for a small fee [38]. The scores were summed to indicate the number of activities for each individual.

Ikigai. Ikigai was measured using the Ikigai-9 scale [39], a measure of the degree to which one feels Ikigai. Higher scores on the Ikigai-9 scale indicate higher Ikigai awareness, a more optimistic outlook on one’s current life and future, and a more positive perception of one’s existence in relation to others and society. Table 1 shows the Ikigai-9 scale as translated by Yoshida et al. [40]. Responses were calculated using a five-point Likert-type scale ranging from “strongly disagree” (1 point) to “strongly agree” (5 points); the total score was calculated by summing the item scores (9–45 points). In a previous study, the reliability and validity of this measure were verified in a survey of 577 general Japanese residents aged 60 years and above; the Cronbach’s alpha was found to be 0.87 [39].

Willingness for New Interactions. The participants were asked the following question: “Would you like to deepen exchanges with people of your generation? This could be answered using the alternatives “1. Not at all”, “2. Not very applicable”, “3. Slightly applicable”, and “4. Very applicable”.

Physical function. Grip strength was used as an index of physical function. Grip strength was measured and calculated according to the 2017 New Physical Fitness Test Implementation Guidelines set forth by the Japanese Ministry of Education, Culture, Sports, Science and Technology [41]. A Smedley grip strength meter was used for measurement. Two measurements were taken on each side, and the average of the higher of the two was used as the score.

### 2.5. Statistical Analysis

The data collected in the baseline and follow-up surveys were used for the statistical analysis. Initially, descriptive statistics for each variable in the baseline and follow-up surveys were calculated to confirm the distribution. Next, a causal model was developed and analyzed to test the hypotheses (Figure 2). We used path analysis for the analysis. Baseline data were used to analyze physical function and social activity, which are considered predictors of Ikigai. Follow-up data were used to determine temporal causality between Ikigai and its predictors. In addition, from the follow-up survey, data on “willingness for new interactions” were used to clarify the relationship with Ikigai. Further, the goodness of fit of the model was calculated by path analysis. IBM Statistical Package for the Social Sciences (SPSS) Statistics 26 and IBM SPSS Amos 25 were used for data analysis, and the significance level was set at 5%.

## 3. Results

The sample consisted of 427 participants in the baseline survey, of which 140 completed the follow-up survey (recovery rate of 32.8%). In selecting the participants for analysis, the exclusion criterion included the Ikigai-9 score outliers, calculated using the quartile range. The lower limit was computed by subtracting the product of the quartile range and 1.5 from the first quartile. Similarly, the upper limit was calculated by adding that amount to the third quartile. In all, two outliers and six individuals with missing items were removed from the analysis. Therefore, the final sample included 132 people, and this value was used as the number of valid responses (30.9% valid response rate).

Table 2 shows the characteristics of the respondents. The mean age was 77.3 ± 5.3 years. Regarding family form, 53 participants (41.1%) lived alone, while 76 (58.9%) resided with their spouses, children/grandchildren, or siblings. With respect to self-rated health, 1 (0.8%) individual reported “not healthy”, 12 (9.2%) indicated “not so healthy”, 109 (83.8%) were “slightly healthy”, and 8 (6.2%) were “very healthy”. The mean grip strength was 20.7 ± 4.2 kg. Regarding social activities, 76 (57.6%) partook in neighborhood association activities, 46 (34.8%) in older person’s clubs, 30 (22.7%) in salons, 41 (31.1%) in volunteer activities, 61 (46.2%) in study sessions, 108 (81.8%) in sports, 107 (81.1%) in hobbies/entertainment, and 101 (76.5%) in travel/vacation. The mean number of social activities was 4.3 ± 1.8. The mean Ikigai-9 score was 29.7 ± 6.3 points. With regard to the responses for willingness for new interactions, 9 (6.8%) subjects reported “not very applicable”, 69 (52.3%) indicated “slightly applicable”, and 54 (40.1%) stated “very applicable”.

We fitted the survey data to a causal model (Figure 2) consisting of four observables that were developed as a hypothesis and assessed. The correlation and path coefficients between the variables in the modified model are shown in Figure 3 and Table 3. No significant path was drawn from physical function to Ikigai (*p* = 0.113). The path coefficients were 0.230 (*p* = 0.006) for social activity to Ikigai, and 0.211 (*p* = 0.014) for Ikigai to willingness for new interactions. This resulted in an indirect and overall effect of 0.049 from social activity to willingness for new interactions.

The analysis indicated that the chi-square value was 1.109 (*p* = 0.574), goodness of fit index was 0.996, adjusted goodness of fit index was 0.979, normed fit index was 0.938, comparative fit index was 1.000, root mean square error of approximation was <0.001, and the Akaike’s information criterion was 17.109, indicating a good model.

## 4. Discussion

The findings indicated that older women’s participation in more social activities resulted in higher Ikigai two years later. Ikigai encompasses motivation and interpersonal satisfaction [15], and PIL, an analogue of Ikigai, includes motivation that guides behavior [42]. Thus, it can be inferred that involvement in a large number of social activities evokes Ikigai. First, we discuss the aspect of behavior being guided by motivation. By participating in numerous social activities, participants perceived that they were in good health [43] and experienced positive emotions, such as higher self-esteem [44]. Their confidence was increased by their ability to take part in events and maintain their health despite their advanced age. Furthermore, self-esteem motivated participation in activities [45], which, in turn, enhanced Ikigai [46]. Positive perceptions, developed through participation in activities, may have fueled the drive for further involvement, evoking Ikigai. Thus, it is suggested that Ikigai is based on positive feelings about oneself, categorized as self-esteem.

The next section discusses the aspect of satisfaction derived from social relationships. Older women have been shown to be desirous of greater activity participation [47,48]. Participating in more activities leads to more opportunities to play a role [49], and being aware of one’s role in activities increases Ikigai [46]. Since participation in social activities is a way to build relationships, it is assumed that women can reaffirm their role recognition by engaging in many activities. These results suggest that the recognition of self-esteem and role recognition through activities may lead to high Ikigai.

In contrast, physical function was found to be unassociated with Ikigai in healthy older women. Poor physical function was related to inadequate subjective mental health and quality of life [50]. Moreover, it is reported to be associated with reduced independence [51]. Based on these factors, it may affect their opportunities to socialize and perform activities of daily living, making people aware of their limitations and affecting their mental health. However, physical function above a certain level was found not to affect them [50]. The participants in this study had the average physical function of healthy older Japanese women; furthermore, their self-rated health was high. Ikigai is a kind of measure of psychological well-being and may reflect a facet of mental health. It is possible that physical functioning was not a factor influencing Ikigai in older adults with a high level of physical health. These findings suggest that Ikigai in older women is more likely to be derived from cognition than through health status.

Ikigai affected participants’ willingness for new interactions with the same generation. Older women have a wider network than men [52] and place a higher value on developing close interpersonal bonds [3]. Thus, good relationships with others are important to older women. However, in old age, there is a loss of social connections and fewer opportunities to meet others, making it challenging to establish new interpersonal relations [53]. This is perhaps because as individuals experience social loss with age, they become more anxious [54,55] and less eager to develop new relationships. However, in this study, Ikigai affected the willingness for new interactions with people from the same generation, suggesting that it may be a source of motivation to establish new interpersonal relationships. In addition, the higher the level of Ikigai, the higher the motivation to engage with others among older women, suggesting that Ikigai may have a function in maintaining the willingness to establish social relationships, which is believed to decline in old age. In the West, the ability to savor positive experiences predicts greater happiness and greater satisfaction with life in older women [56]. Similarly, the present study indicates that motivation to live may be associated with positive behavior, and this behavior may lead to greater happiness for older women. From a practical point of view, the examination of interventions that enhance life satisfaction among older women will lead to the enhancement of well-being of older women throughout the world, not only in Asian countries.

Women have a longer life expectancy than men and are more likely to experience isolation and separation from family and friends [54,57]. Finding value in relationships and staying connected with others is a desirable way for them to age and enhance their social health. Ikigai was found to have the potential to provide a desirable way to maintain interpersonal relationships and support their social health.

### Study Limitations

As this study used data from the same time period for Ikigai and “willingness for new interactions”, it was not possible to establish a temporal causal relationship. However, we were able to show the possibility of an association between Ikigai and social aspects, which has not been tested in previous studies. It will be necessary to conduct future research to construct a model with reference to the relationship between Ikigai and social aspects. In addition, the development and implementation of Ikigai-focused interventions can be expected to improve the social health of older women, such as through building new connections. For this purpose, it is necessary to clarify causality in the relationship between Ikigai and social aspects through longitudinal studies that utilize follow-ups.

This study has several limitations. First, the perception and behavior of social aspects were measured using only psychological factors of motivation. The item “willingness for new interactions” alone is not representative of the social aspect. More items need to be added to see if Ikigai contributes to the social aspect. Moreover, the impact of Ikigai on actual behaviors could not be examined and the findings were based on the responses of the study’s participants. Future research should focus on actual behaviors related to forming new interpersonal relationships to determine whether Ikigai influences them, based on the current results. In addition, it should be noted that “willingness for new interactions” showed a bias in the responses. As this variable was not normally distributed, reliability and validity need further examination. Further, the mediating effect of Ikigai should be examined and clarified in future research.

Second, the variable of social activity was assessed in terms of the number of interpersonal activities that the respondents engaged in; however, the frequency of engaging in these activities was not considered. The negative effects of frequent participation in such activities have been previously reported [58]. Therefore, it is necessary to consider this factor in the future.

## 5. Conclusions

In this study, we longitudinally examined the relationship between Ikigai and its predictors. We also tested whether Ikigai has an effect on willingness for new interactions as a model. The results of the analysis showed that participating in many social activities contributed to having high Ikigai. In addition, Ikigai had a significant effect on willingness for new interactions. This finding indicates the usefulness of Ikigai in maintaining the social health of older women. It will be necessary to consider Ikigai as one of the means by which older women can attain social health.

## Figures and Tables

**Figure 1 ijerph-18-08718-f001:**
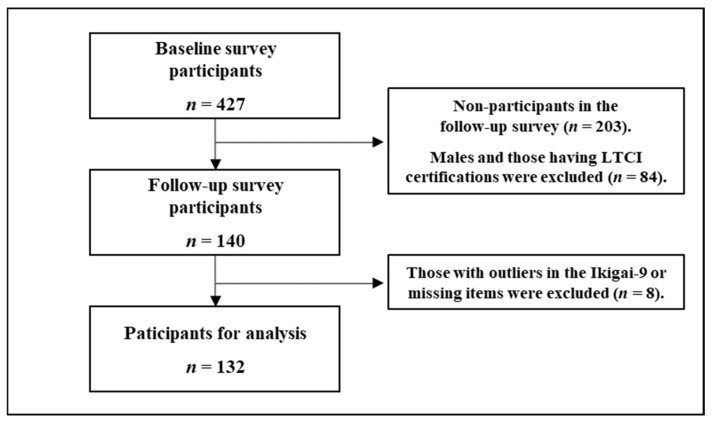
Flow chart presenting the selection procedure of the study participants.

**Figure 2 ijerph-18-08718-f002:**
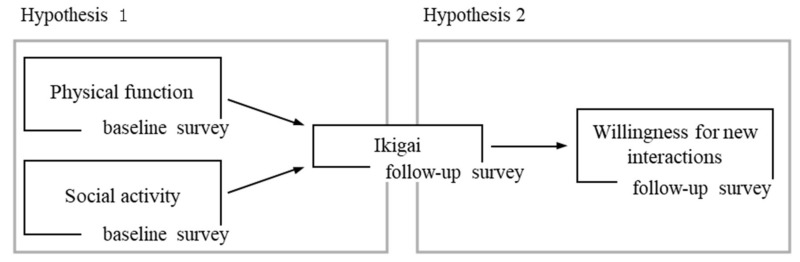
A causal model of predictors (Hypothesis 1) and effects (Hypothesis 2) of Ikidai.

**Figure 3 ijerph-18-08718-f003:**
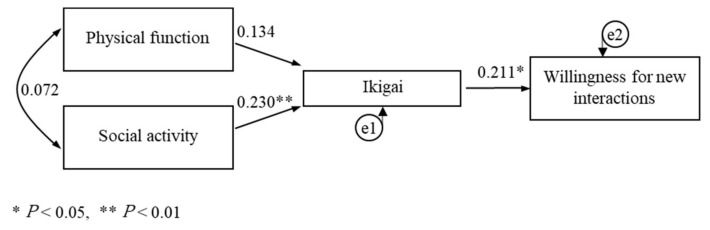
A modified model of predictors and effects of Ikigai.

**Table 1 ijerph-18-08718-t001:** Items of Ikigai-9.

1	I often feel that I am happy.
2	I want to learn something new or start.
3	I think that I am useful for something else or society
4	I am relaxed mentally
5	I am interested in various things.
6	I think that my existence is necessary for something or someone else.
7	My life is abundant and fulfilling.
8	I want to extend my possibilities.
9	I think that I am influencing someone.

Yoshida, I., Hirao, K., and Kobayashi, R. (2019). The effect on subjective quality of life of occupational therapy based on adjusting the challenge–skill balance: a randomized controlled trial. Clinical Rehabilitation, 33(11), 1732–1746.

**Table 2 ijerph-18-08718-t002:** Characteristics of the study participants. (*N* = 132).

Baseline Survey		Mean ± *SD* *n* (%)	Range
Age		77.3 ± 5.3	65–90
Family form			
	Living alone	53 (41.1)	
	Living together	76 (58.9)	
Self-rated health			
	1. Not healthy	1 (0.8)	
	2. Not so healthy	12 (9.2)	
	3. Slightly healthy	109 (83.8)	
	4. Very healthy	8 (6.2)	
Social activity ^(^ª⁾			
	Salon	30 (22.7)	
	Older person’s club	46 (34.8)	
	Neighborhood association	76 (57.6)	
	Study session	61(46.2)	
	Volunteer activities	41 (31.1)	
	Sports	108 (81.8)	
	Hobby/Entertainment	107 (81.1)	
	Travel/Vacation	101 (76.5)	
	Total number of activities	4.3 ± 1.8	
Grip strength (kg)		20.7 ± 4.2	7.8~36.7
**Follow-Up Survey**		**Mean ±** ***SD*** ***n* (%)**	**Range**
Willingness for new interactions			
	1. Not at all	0 (0.0)	
	2. Not very applicable	9 (6.8)	
	3. Slightly applicable	69 (52.3)	
	4. Very applicable	54 (40.1)	
Ikigai-9		29.7 ± 6.3	9~45

^(^ª⁾ For the social activities dimension, multiple answers were allowed.

**Table 3 ijerph-18-08718-t003:** Correlation and path coefficients of the modified model (*N* = 132).

Coefficients between Variables (Correlation)		Correlation Coefficient			*p*
Physical function		0.072			0.413
Social activity					
Coefficients between variables (Causal)		Direct effect	Indirect effect	Overall effect	
Exogenous variables	Endogenous variables				
Social activity	Ikigai	0.230	–	0.230	0.006
Social activity	Willingness for new interactions	–	0.049	0.049	–
Ikigai	Willingness for new interactions	0.211	–	0.211	0.014

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
