# Peer review of "Predictors and Importance of Social Aspects in Ikigai among Older Women"

_ijerph, 2021, doi:10.3390/ijerph18168718_

Round 1

Reviewer 1 Report

Overall, I enjoy reading this manuscript and learning the concept of Ikigai. Ikigai is a significant idea. I would suggest the authors to introduce this concept more broadly and precisely. I learned the whole picture after searching for more information about Ikigai by myself.

  1. A figure for Ikigai might be helpful for readers to catch the main framework.
  2. I can see many Western traditional or new theories related to Ikigai, such as broaden and build theory, Cohen's main effect model/buffer model and others from the filed of positive psychology. A paragraph to compare these Western and Japan theories to emphasize the significance of Ikiga for Asian older women is needed. Moreover, it could emphasize the contribution of this study.  In addition, the author might also stress whether the practice if Ikiga can be examined in other countries.
  3. More information for the Japanese aging issue, especially older women, is needed.
  4. Measures: more information or previous studies to support why these items were included in the social activity measurement.
  5. Since this is a longitudinal study, why the outcome variable is "willingness for new interactions", but not the actual frequency of social interaction?
  6. A timeline added in Figure 2 could be helpful.
  7. Many other personal characteristics can affect the hypotheses, such as the number of clinical diseases. Did the author collect this information?
  8. Mediating effect of Ikigai should be examined as well.

Author Response

Thank you for the opportunity to submit the revised draft of our manuscript titled “Predictors and importance of social aspects in Ikigai among older women” to the International Journal of Environmental Research and Public Health. We appreciate the time and effort that you have dedicated to providing valuable feedback on our manuscript. We are grateful to the reviewers for their insightful comments on our paper and have incorporated changes to reflect the suggestions provided by them.

Reviewer 2 Report

I recommend the publication of this article with the following minor revisions:

  • the study participants inclusion and exclusion criteria should be better explained and justified;
  • The measure procedures should be better grounded. Considering that physical function was determined as a central variable of the study, further explain how it was assessed; better substantiate the strategy used to evaluate social activity;
  • Explicit the statistical analysis procedures used.

Author Response

Thank you for the opportunity to submit the revised draft of our manuscript entitled “Predictors and importance of social aspects in Ikigai among older women” to International Journal of Environmental Research and Public Health. We appreciate the time and effort that you have dedicated to providing valuable feedback on our manuscript. We are grateful to the reviewers for their insightful comments on our paper and have been able to incorporate changes to reflect most of the suggestions provided by them.

Round 2

Reviewer 1 Report

I am satisfied with the revised manuscript.